# Regulation of Plant Photoresponses by Protein Kinase Activity of Phytochrome A

**DOI:** 10.3390/ijms24032110

**Published:** 2023-01-20

**Authors:** Da-Min Choi, Seong-Hyeon Kim, Yun-Jeong Han, Jeong-Il Kim

**Affiliations:** 1Department of Integrative Food, Bioscience and Biotechnology, Chonnam National University, Gwangju 61186, Republic of Korea; 2Kumho Life Science Laboratory, Chonnam National University, Gwangju 61186, Republic of Korea

**Keywords:** phytochrome A, phosphorylation, protein kinase, phytochrome-interacting factors, plant photoresponses

## Abstract

Extensive research has been conducted for decades to elucidate the molecular and regulatory mechanisms for phytochrome-mediated light signaling in plants. As a result, tens of downstream signaling components that physically interact with phytochromes are identified, among which negative transcription factors for photomorphogenesis, PHYTOCHROME-INTERACTING FACTORs (PIFs), are well known to be regulated by phytochromes. In addition, phytochromes are also shown to inactivate an important E3 ligase complex consisting of CONSTITUTIVELY PHOTOMORPHOGENIC 1 (COP1) and SUPPRESSORs OF *phyA-105* (SPAs). This inactivation induces the accumulation of positive transcription factors for plant photomorphogenesis, such as ELONGATED HYPOCOTYL 5 (HY5). Although many downstream components of phytochrome signaling have been studied thus far, it is not fully elucidated which intrinsic activity of phytochromes is necessary for the regulation of these components. It should be noted that phytochromes are autophosphorylating protein kinases. Recently, the protein kinase activity of phytochrome A (phyA) has shown to be important for its function in plant light signaling using *Avena sativa* phyA mutants with reduced or increased kinase activity. In this review, we highlight the function of phyA as a protein kinase to explain the regulation of plant photoresponses by phyA.

## 1. Introduction

In higher plants, light is an important environmental cue to optimize their growth and development, causing the evolution of multiple and sophisticated photoreceptor systems for sensing continually changing surroundings [1]. There are more than 10 photoreceptors found in *Arabidopsis thaliana* L. Heynh, including UV RESISTANCE LOCUS 8 (UVR8) for sensing UV-B (280–320 nm), UV-A/blue (320–500 nm) light-sensing photoreceptors such as phototropins, cryptochromes, Zeitlupes (ZTL/FKF1/LKP2), and red/far-red (600–750 nm) light-sensing phytochromes [2,3]. Among them, phytochromes are the most extensively studied thus far [4]. They are dimeric chromoproteins with each monomer possessing a covalently linked chromophore (i.e., phytochromobilin), and exist in two photo-reversible forms, red (R, 660 nm) light-absorbing Pr and far-red (FR, 730 nm) light-absorbing Pfr forms [5]. In general, the Pfr form is considered as the biologically-active form of phytochromes. In Arabidopsis, phytochromes are encoded by a family of five genes, *PHYA* to *PHYE* [6,7], where the chromophore-assembled phytochrome A (phyA) is classified as a light-labile type Ⅰ phytochrome and where phyB-phyE belong to relatively light-stable type Ⅱ phytochromes [8]. In dark-grown plants, phyA is the most abundant phytochrome, but the level of phyA reduces rapidly upon formation of the Pfr form through exposure to light. Thus, type Ⅱ phytochromes are abundant in light-grown plants where phyB becomes the most abundant. Due to the difference in light-dependent protein stability between phyA and type II phytochromes, phyA plays a major role in plant development during the transition from dark to light; whereas type Ⅱ phytochromes play prominent roles in light-grown plants [9,10].

In addition to the light-labile property, phyA is unique in mediating plant responses to FR light, whereas other phytochromes regulate plant responses to R light [11]. FR light is favorable for generating the inactive Pr form so how phyA functions in FR light had been an unresolved question for a long time. A joint experimental-theoretical study answered this question by proposing the nucleocytoplasmic shuttling of phyA via FAR-RED ELONGATED HYPOCOTYL 1 (FHY1) and FHY1-LIKE (FHL) as a decisive role for the FR-light signaling [12]. Due to the partial overlap of absorption spectra between the Pr and Pfr forms, FR light is able to transform a small proportion of Pr into Pfr. With Pfr formation, FHY1 and FHL function as shuttle proteins for the nuclear import of phyA, binding to the Pfr form of phyA in the cytosol, and transporting it into the nucleus and back to the cytosol when phyA is converted to Pr under the FR light [13,14]. Thus, photocycling between the Pr and Pfr forms of phyA and the Pfr-specific interaction with FHY1/FHL is essential for the phyA function under FR light, especially mediating FR-high irradiance responses (FR-HIRs) in plants [15]. Moreover, phyA signaling is initiated by very low light amounts of any wavelength in which tiny amounts of the Pfr form of phyA can be generated, mediating very low fluence responses (VLFRs) in plants. Therefore, phyA detects FR-enriched light conditions, i.e., close vegetation or canopy shade, to mediate FR-HIRs for survival, and induces VLFRs for germination and de-etiolation [9,11,16].

Plants in the dark or soil after seed germination exhibit skotomorphogenic development, i.e., etiolated seedlings with elongated hypocotyls and folded cotyledons with hook, which, upon exposure to light, show photomorphogenic development, i.e., de-etiolated seedlings with opened cotyledons and chlorophyll biosynthesis [17]. The transition from skotomorphogenesis to photomorphogenesis is essential for the successful establishment of seedlings, which requires a highly regulated signaling network for the transcription of photoresponsive genes through the concerted work of photoreceptors, E3 ubiquitin ligases, and various transcription factors [18,19,20]. Although phytochromes are known as one of the most important photoreceptors for the regulation of plant photomorphogenesis [21,22], the molecular mechanisms for the regulation of downstream signaling components by phytochromes have not been fully elucidated. In this review, we highlight current knowledge about the molecular and regulatory mechanisms of phyA in plant light signaling by focusing on the fact that phytochromes are autophosphorylating protein kinases.

## 2. The Core Phytochrome Signaling Pathway

Since the first discovery in 1950s [23,24], phytochromes have been studied intensively by a broad range of experimental approaches, and a representative model about how phytochromes transduce light signals into plant physiological responses can be proposed (Figure 1). In the dark, phytochromes are biosynthesized as the inactive Pr form and accumulated in the cytoplasm, and upon exposure to light, the Pr form is phototransformed into the active Pfr form. Thus, the first step for the phytochrome signaling is the conformational changes that are triggered by absorbing the incoming light including R and FR wavelengths. Then, the second step is the translocation of the photoactivated phytochromes from the cytoplasm into the nucleus, which is suggested as a crucial control step for the phytochrome signaling [14,25,26]. In the nucleus, the third step is the interaction of phytochromes with various downstream signaling components, such as PHYTOCHROME-INTERACTING FACTORs (PIFs) and an E3 ubiquitin ligase complex of CONSTITUTIVELY PHOTOMORPHOGENIC 1 (COP1) and SUPPRESSORs OF *phyA-105* (SPAs) [17,27]. It is notable that both PIFs and COP1/SPA complex play roles for sustaining skotomorphogenesis, resulting in the suppression of photomorphogenesis in the dark. Thus, the inhibition of PIFs and COP1/SPA complex by the photoactivated phytochromes induces the photomorphogenic development. The regulatory mechanisms of PIFs by phytochromes have been identified as ubiquitin/26S proteasome-mediated proteolysis, inhibition of PIFs binding to target promoters (i.e., sequestration), and retention in the cytoplasm (i.e., compartmentation) [28,29,30,31,32]. At the same time, phytochromes in the nucleus inhibit the E3 ligase activity of the COP1/SPA complex by inducing its dissociation [33,34]. Moreover, phytochromes mediate translocation of COP1 from the nucleus to the cytoplasm (i.e., compartmentation) under prolonged light exposure [35,36]. The COP1/SPA complex is a well-studied E3 ligase that induces the 26S proteasome-mediated degradation of many signaling components, such as ELONGATED HYPOCOTYL 5 (HY5), HY5-HOMOLOG (HYH), LONG HYPOCOTYL IN FAR-RED 1 (HFR1), LONG AFTER FAR-RED LIGHT 1 (LAF1), HECATEs (HECs), and B-box containing proteins (BBXs) [37,38,39,40,41,42,43,44]. Among the target proteins of COP1/SPA complex, HY5 is a member of the basic leucine zipper (bZIP) gene family that has been considered as a master transcriptional factor for the expression of photoresponsive genes to initiate photomorphogenesis in plants [19,45]. Therefore, the transcriptional regulation of photoresponsive genes via positive transcriptional factors such as HY5 might be the core regulatory mechanism of phytochrome signaling pathway, which is accomplished by the suppression of negative regulators for photomorphogenesis such as PIFs and COP1/SPA complex (Figure 1).

Phytochromes are involved in almost every step of the plant life cycle, starting from seed germination to seedling de-etiolation, reproductive transition, and senescence [17]. For this, phytochromes regulate various signaling partners by protein-protein interactions [46]. Among them, PIFs are suggested to play pivotal roles as signaling hubs for the function of phytochromes [47]. PIFs belong to the basic helix-loop-helix (bHLH) family of transcription factors, and eight PIFs have been identified in Arabidopsis [48,49,50]. Through the phytochrome-PIF signaling modules, various plant growth and development could be regulated (Figure 2).

As examples, PIF1 plays a critical role in inhibiting light-dependent seed germination, so phytochromes can promote seed germination by the negative regulation of PIF1 [51]. PIF3 primarily functions as a negative regulator of seedling de-etiolation along with other PIFs, so phytochromes induce seedling de-etiolation by inhibiting PIF3 and other PIFs [52,53]. PIF4 and PIF7 are positive regulators of plant thermomorphogenesis, in which phytochromes have been shown to function as thermo-sensors and negatively regulate these PIFs [54,55,56,57]. Therefore, the phytochrome-PIF signaling modules play important roles in the regulation of various plant development, such as photomorphogenesis and thermomorphogenesis [31,49]. It should be noted that Arabidopsis PIFs (PIF1-PIF8) have binding domains to interact with phytochromes, known as active phyB-binding (APB) and phyA-binding (APA) motifs [58]. All the Arabidopsis PIFs have an APB motif, so they can interact with phyB in a light-dependent manner. In contrast, only PIF1 and PIF3 have an APA motif in addition to APB, so these two PIFs can interact with phyA as well as phyB (Figure 2). However, many phytochrome-interacting proteins, other than PIFs, do not have the APB and APA motifs, so these motifs would be one of the binding motifs to phytochromes. In this regard, it is also notable that PIF4, PIF7, and PIF8 can interact with phyA to a lesser strength compared with phyB, although they do not have an APA motif [48,59]. In general, PIF3 and PIF1 are used for the study of phyA. With these backgrounds, we move on the molecular function of phyA as an autophosphorylating protein kinase in the following.

**Figure 2 ijms-24-02110-f002:**
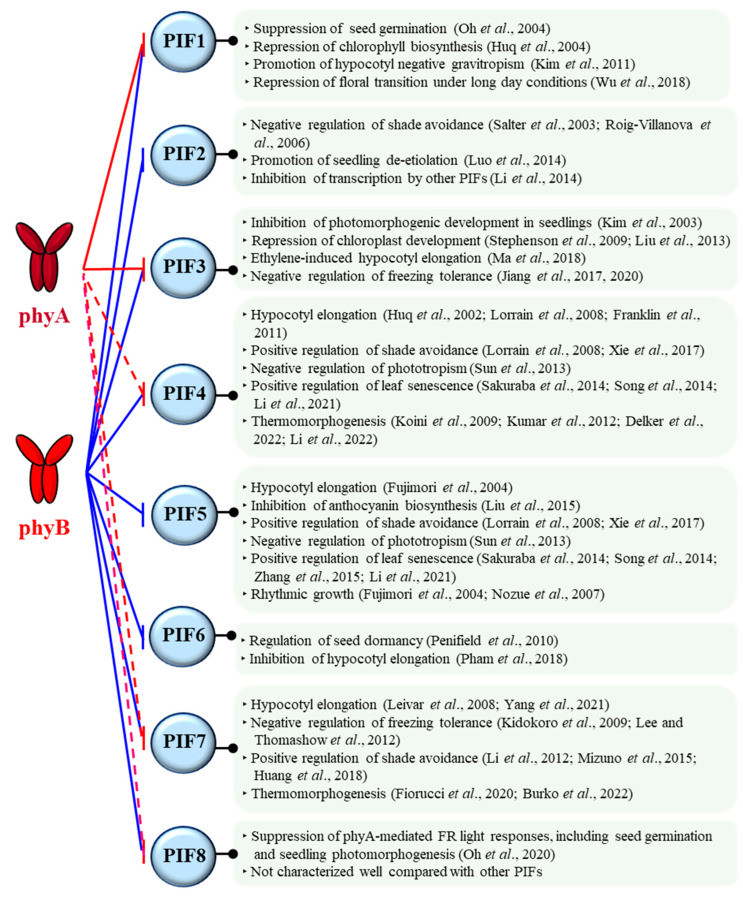
The functions of eight PIFs identified in Arabidopsis. In general, photoactivated phytochromes negatively regulate PIFs by molecular mechanisms such as phosphorylation and subsequent proteolysis, sequestration, and compartmentation. Arabidopsis phyB interacts with all eight PIFs that have an active phyB-binding (APB) motif (shown as blue lines). In contrast, only PIF1 and PIF3 have an active phyA-binding (APA) motif, showing interaction with phyA (red lines). However, it is noted that APA-lacking PIF4, PIF7, and PIF8 can also interact with phyA, although their binding affinities to phyA are much lesser than that to phyB (red dashed lines) [29,48,49,51,52,53,56,57,59,60,61,62,63,64,65,66,67,68,69,70,71,72,73,74,75,76,77,78,79,80,81,82,83,84,85,86,87,88,89,90,91,92].

## 3. Regulation of phyA as Phosphoproteins

Plant phytochromes have been reported as phosphoproteins by a phosphate analysis with purified phytochromes from dark-grown *Avena sativa* L. (oat) seedlings, indicating an average of one phosphate per monomer [93]. Phytochromes exist as dimers, and the molecular mass of each monomer is about 120–130 kDa. They consist of a globular chromophore-binding N-terminal domain and a structurally extended C-terminal domain, also known as photosensory module (PSM) and output module (OPM), respectively (Figure 3) [4,94]. The PSM contains the chromophore-harboring photosensory core (i.e., PAS-GAF-PHY tri-domain) and N-terminal extension (NTE), in which the NTE is shown to be dispensable for chromophore binding but necessary for the biological activity [95,96,97,98]. The OPM contains a PAS-repeat domain (PRD) and a histidine kinase-related domain (HKRD) that play roles in phytochrome function such as dimerization, nuclear localization, and interactions with signaling partners [99,100]. Phytochromes were shown to be readily phosphorylated in vitro and in vivo [101,102], and to this end, three phosphorylation sites on *A. sativa* phyA (AsphyA) have been identified: two phosphorylation sites (Ser-8 and Ser-18) in the NTE, and one site (Ser-599) in the hinge region between PSM and OPM [103]. The determination of these phosphorylation sites subsequently opened the functional analysis of phyA as phosphoproteins.

Based on the observation that Ser-599 in the hinge region of AsphyA is phosphorylated in vitro and in vivo in a Pfr-specific manner [103], the functional role of Ser-599 phosphorylation was first investigated using transgenic plants with S599A mutant. The results showed hypersensitive responses of the S599A plants to FR light, suggesting an inhibitory role of the phosphorylation in the hinge region of AsphyA [104]. The same study showed that the S599A mutant (i.e., unphosphorylated form) of AsphyA interacted more strongly with nucleoside diphosphate kinase 2 (NDPK2), a positive signaling component in the phytochrome signaling [105]. Thus, these results suggest that the hinge region phosphorylation plays a role for the regulation of protein-protein interactions between phytochromes and signaling partners. To be consistent, three sites (Ser-590, Thr-593, and Ser-602) in the hinge region of *A. thaliana* phyA (AtphyA) have been demonstrated to be phosphorylated in plants, indicating that the phosphorylation at hinge region plays an important role in regulating the function of AtphyA [106]. In the same study, it was also shown that the phosphorylated form of AtphyA is produced in the nucleus by the Pfr form, which proposes the role of phytochrome phosphorylation in the nucleus.

Subsequently, the Ser-8 and Ser-18 in the NTE have been identified as the autophosphorylation sites of AsphyA, and the Ser-to-Ala autophosphorylation site mutants (ex. S8/18A) were degraded in plants at a significantly slower rate than wild-type AsphyA, resulting in transgenic plants with hypersensitive responses to FR light [107]. These results are consistent with an older report that the serine-to-alanine substitutions in the NTE including Ser-8 and Ser-18 increased the biological activity [108]. Based on these data, the autophosphorylation of AsphyA is suggested as a desensitization signal to terminate phyA signaling after its activation under light, which might be comparable to rhodopsin desensitization via phosphorylation by rhodopsin kinase in animals [109]. It is notable that phyA exhibits rapid light-induced proteolytic degradation along with its aggregation, known as sequestered areas of phytochrome (SAPs), which requires the ubiquitin/26S proteasome pathway [110,111]. In addition, it is reported that the light-induced degradation of phyA is promoted by transfer into the nucleus, although phyA is degraded both in the cytoplasm and in the nucleus [112]. Thus, phyA mediates only transient signaling under R light because of its rapid degradation, while phyA functions stably under FR light or shade, i.e., in the conditions that its degradation is prevented or significantly reduced [113]. Furthermore, COP1 has been reported as an E3 ligase for the ubiquitination of phyA, which associates preferentially with the phosphorylated form of phyA [114,115]. Collectively, the autophosphorylation in the NTE might play an inhibitory role in the phyA signaling by causing the rapid light-induced degradation (Figure 3). By comparison, phosphorylation sites of AtphyB in the NTE, such as Ser-86 and Tyr-104, have also been reported, and the NTE phosphorylation is suggested to play a negative role in phyB function [116,117]. Especially, the phosphorylation at Ser-86 has shown to accelerate dark reversion (i.e., light-independent thermal conversion of Pfr to Pr), which contribute to the inactivation of type II phytochromes (phyB, phyD, and phyE) in plants [116,118]. Therefore, the phosphorylation of phyA and phyB in the NTE might provide a mechanism of signal desensitization or attenuation of phytochromes: for phyA via promoting protein degradation and for phyB via accelerating dark reversion.

Since phytochromes are phosphoproteins, it is expected that protein phosphatases might be involved in the phytochrome signaling. Indeed, a few protein phosphatases that directly interact with dephosphorylate phytochromes has been reported, which include flower-specific phytochrome-associated protein phosphatase (FyPP), phytochrome-associated protein phosphatase 5 (PAPP5), and phytochrome-associated protein phosphatase 2C (PAPP2C) [119,120,121]. Especially, the protein stability of phyA increased in PAPP5-overexpressing plant, but decreased in *papp5* mutant, suggesting that phyA phosphorylation is necessary for the promotion of protein degradation [121]. In addition to PAPP5 that can increase phyA stability in plants, both FyPP and PAPP2C also play positive roles in the phytochrome signaling, suggesting a positive regulation through protein dephosphorylation. These results are consistent with the role of phyA autophosphorylation that negatively regulates phytochrome signaling. Therefore, phyA signaling could be regulated by reversible phosphorylation, in which autophosphorylation decreases signaling flux by reducing the amounts of active Pfr and dephosphorylation increases the signaling flux by increasing protein stability and/or regulating the protein-protein interactions with signaling partners (Figure 3).

## 4. Function of phyA as Protein Kinases

A protein can be phosphorylated by itself (i.e., autophosphorylation) or by protein kinases (PKs). In this regard, it is notable that earlier studies on phytochromes raised issues about whether phytochromes are protein kinases or not [122]. This is based on the results that the phosphorylation of nuclear proteins involved in phytochrome signaling and sequence analyses indicated similarity between the HKRD of phytochromes and prokaryotic histidine kinases [123,124]. In addition, a cyanobacterial phytochrome (Cph1) has been reported as a light-regulated histidine kinase showing autophosphorylation [125]. However, it is suggested that the HKRD in plant phytochromes is not a functional domain, because the key conserved residues of the histidine kinase domain are absent and mutations on residues required for ATP-binding did not affect the function of phytochromes [122]. Later, not only phytochromes isolated from dark-grown seedlings, but also purified recombinant phytochromes were proven to exhibit serine/threonine kinase activity, suggesting that eukaryotic phytochromes are histidine kinase paralogs with serine/threonine specificity [126]. To this end, recombinant phytochromes isolated from different plant species have all shown autophosphorylation and the kinase activity on histone H1 as a substrate, demonstrating that plant phytochromes are protein kinases [127]. Moreover, since the HKRD is not a functional kinase domain, it is conceivable that the kinase domain in plant phytochromes resides in the domain other than the HKRD. This issue has been answered by kinase domain mapping experiments, which suggests the N-terminal photosensory core composed of PAS-GAF-PHY tri-domain as the domain for the observed kinase activity of AsphyA [127]. Additionally, a previous report had predicted nucleotide binding sites of phytochromes in the photosensory core [128]. In addition, the tertiary structure of Cph1 photosensory core is suggested to be similar to the regulatory region of cyclic nucleotide phosphodiesterases and adenylyl cyclases, indicating the possibility of ATP-binding in the region [129]. Moreover, the HKRD was reported to play roles in phytochrome dimerization and protein-protein interactions with signaling partners, such as PAPP5 [100,121]. Therefore, plant phytochromes are now believed to be the autophosphorylating serine/threonine kinases, and the PAS-GAF-PHY tri-domain, not the HKRD, is responsible for the observed kinase activity.

After the proposal of plant phytochromes as serine/threonine kinases [126], several substrate proteins that can be phosphorylated by phytochromes have been reported, which includes PHYTOCHROME KINASE SUBSTRATE 1 (PKS1), cryptochromes, Aux/IAA proteins, FHY1, and PIFs [28,71,91,130,131,132,133,134,135]. Among these substrates, PIFs are remarkable in showing phytochrome-dependent phosphorylation in plants, which is required for their degradation via the ubiquitin/26S proteasome pathway. It is notable that the regulation of PIF7 by phosphorylation is different from other PIFs, which is necessary for controlling its localization, not for proteolysis [56,91]. At the end of these substrate studies, phytochromes have been demonstrated to be the protein kinases for the phosphorylation of PIFs [127]. It should be noted that the autophosphorylation of AsphyA is reduced only in the presence of PIF3, but not in the presence of PKS1 and other known substrate candidates, suggesting PIFs as the genuine substrates of phytochrome kinase activity. Subsequently, the functional roles of phyA kinase activity have been suggested by analyzing AsphyA mutants with altered kinase activity [127,136]. Three AsphyA mutants displaying reduced kinase activity (K411L, T418D, and D422R) were initially isolated by proteomics and site-directed mutagenesis on the photosensory core, in which the ATP-binding affinity of the mutants was significantly decreased compared with wild-type AsphyA. As a result, transgenic plants of the AsphyA mutants exhibited hyposensitive responses to FR light. Moreover, FR-induced phosphorylation and subsequent degradation of PIF3 was significantly decreased in the transgenic plants of the AsphyA mutants with reduced kinase activity. This study provides the evidence that AsphyA functions as a protein kinase in triggering PIF3 degradation via phosphorylation in plants [127]. Later, two AsphyA mutants displaying increased kinase activity (K411R and T418V) were further isolated and their function has been investigated in Arabidopsis. The results showed that these mutants accelerated FR-induced phosphorylation and subsequent degradation of both PIF3 and PIF1 in transgenic plants, exhibiting hypersensitive de-etiolation responses to FR and higher germination frequencies than the transgenic plant with wild-type AsphyA [136]. The same study also showed that HY5 accumulated higher in the transgenic plants under FR light than in the control plants. These results suggest that AsphyA mutants with increased kinase activity inhibited the activity of COP1/SPA complex more effectively, compared with wild-type AsphyA, for the accumulation of HY5. Overall, these studies suggest a positive relationship between the protein kinase activity of AsphyA and photoresponses in plants (Figure 4).

Besides phytochromes, PIFs have been reported to be phosphorylated by other PKs, which include casein kinase II (CK2), brassinosteroid insensitivity 2 (BIN2), light-regulated protein kinases (PPK1 to PPK4), and SPAs [137,138,139,140,141]. In plants, slow-migrating PIF proteins are detected, probably due to multiple phosphorylations, but phytochromes alone are not enough to mediate the hyperphosphorylation of PIF3 in plants [127]. Thus, co-action of phytochromes and other PKs might be necessary for the regulation of PIF function by phosphorylation. However, the phosphorylation of PIFs is not observed in the absence of phytochromes, indicating phytochromes as the most important PK for PIFs.

Besides PIFs, several signaling components in the core phytochrome signaling pathway are known to be regulated by phosphorylation. A previous report suggests phyA-mediated phosphorylation of FHY1 in a R and FR light-reversible manner [133,142]. The study showed R light-induced phosphorylation of FHY1, which can be reversed by the exposure to FR light. It is well known that the nuclear localization of phyA depends on a nuclear localization signal (NLS) of FHY1, which is recognized by importing α (IMPα) independently of phyA [26]. FHY1 is phosphorylated on serine residues close to the NLS, resulting in the prevention of FHY1 binding to IMPα. Thus, the Pfr form of phyA might induce the phosphorylation of FHY1 to accelerate the dissociation of FHY1-IMPα, which helps FHY1 back into the cytoplasm for another cycle of phyA shuttle to the nucleus. Thus, FHY1 and FHL might be another candidate for substrates of phyA kinase activity. However, AsphyA mutants with altered kinase activity did not show significant differences in the nuclear localization [127,136]. Thus, the importance of R-light induced phosphorylation of FHY1 in phyA signaling needs to be investigated in the future. The other signaling component that can be regulated by phosphorylation is COP1. A serine/threonine kinase PINOID (PID) has been reported to repress the activity of COP1 by phosphorylating at Ser-20, resulting in the promotion of photomorphogenesis [143]. This study suggests that COP1 phosphorylation is necessary for the dissociation of the COP1/SPA complex and possibly for the nuclear exclusion of COP1. Considering the direct interaction of phyA with COP1, phyA is also possible to involve in the phosphorylation of COP1. Alternatively, it is also notable that COP1-interacting SPA1 is reported to act as a serine/threonine kinase [140]. Therefore, further studies are necessary to identify the protein kinase(s) for COP1 and to elucidate the functional roles of COP1 phosphorylation.

## 5. The Regulatory and Molecular Mechanisms of phyA in Plant Light Signaling

Phytochrome-mediated light signaling in plants is ultimately accomplished via the expression of photoresponsive genes for photomorphogenesis. An initial genome-wide expression analysis of *A. thaliana* has shown that approximately 2500 genes were regulated by phytochromes upon exposure to light, where ~80% of the photoresponsive genes are induced and ~20% are repressed [144]. This study also showed that phyA has a dominant role in the light-induced expression of early response genes. The majority of these genes are related to the transition from heterotrophic to autotrophic life of plants [145,146]. For the transcriptional regulation of thousands of genes, master regulatory transcription factors (TFs) are necessary to bridge the phytochrome activation with transcriptional reprogramming. Basically, the regulation of the photoresponsive genes by phytochromes might be dependent on PIFs and HY5. Because PIFs are known to sustain skotomorphogenesis, photoactivated phyA can inhibit the expression of genes for skotomorphogenesis by inactivating PIFs [49,147]. At the same time, phyA induces the accumulation of HY5 by inactivating the COP1/SPA complex, which induces the expression of the photoresponsive genes [19,45,148,149]. Thus, plant phytochromes exert their function by regulating the expression and accumulation of HY5, a master regulator of photomorphogenesis including de-etiolation. For example, phyA is not detectable in the nucleus of dark-grown seedlings but accumulates in the nucleus within minutes following exposure of seedlings to light, resulting in HY5 accumulation in higher amounts. Therefore, the primary regulatory mechanism of phyA for photomorphogenesis would be the control of HY5 levels in plants.

With the above advances, the molecular mechanisms of phyA in plant light signaling can be speculated (Figure 5). In the dark, phyA is biosynthesized as the Pr form (PrA) and accumulates in the cytoplasm. In the nucleus, PIFs regulate gene expression for sustaining skotomorphogenesis, and the COP1/SPA complex degrades HY5 to inhibit photomorphogenesis. Upon exposure to FR light, the Pr-to-Pfr photoactivation occurs and the photoactivated phyA (PfrA) translocalizes from the cytoplasm into the nucleus via FHY1/FHL shuttle. Under continuous FR light, the Pfr form in the nucleus is transformed into the Pr form, which dissociates the FHY1/FHL-phyA complex. It is thought that the phosphorylation of FHY1/FHL might accelerate this dissociation, which can be mediated by phyA and/or possibly by other PKs. In the nucleus, phyA induces the degradation of PIFs by directly phosphorylating them as a protein kinase. PIFs can also be phosphorylated by other PKs such as PPKs and SPAs. At the same time, phyA inactivates COP1/SPAs by dissociating the complex. COP1 can be phosphorylated by PID and possibly by phyA, which promotes the dissociation as well as the translocation of COP1 into the cytoplasm. Therefore, the negative regulation of both PIFs and the COP1/SPA complex via protein-protein interactions and kinase activity would be the primary molecular mechanisms of phyA, which induces the accumulation of HY5 for transcriptional regulation to initiate photomorphogenic development. Lastly, phyA is autophosphorylated (AutoP), and the phosphorylated phyA degrades rapidly to desensitize signaling for next round of signal perception.

## 6. Conclusions and Perspectives

Although it is a long-lasting issue whether phytochromes are protein kinases or not, recent advances provide more confident evidence for the issue than before. First, phytochromes are regulated by reversible phosphorylation. As an example, AsphyA is phosphorylated at Ser-8 and Ser-18 in the NTE by autophosphorylation and at Ser-599 in the hinge region. The NTE phosphorylation induces rapid degradation of phyA, playing roles as a desensitization signal, whereas the hinge region phosphorylation regulates the protein-protein interactions with downstream signaling partners (Figure 3). A few protein phosphatases that interact and dephosphorylate phytochromes play positive roles in plant light signaling. All these results are consistent with the old notion that phytochromes are phosphoproteins. However, there are no known PKs that phosphorylate phyA thus far, so the hinge region phosphorylation needs to be investigated further by identifying the PKs. In this regard, it is notable that a positive regulator of phyA signaling, TANDEM ZINC-FINGER/PLUS3 (TZP), is necessary for the formation of phosphorylated forms of phyA in the nucleus under FR light [150,151]. Thus, it is possible that the hinge region phosphorylation might be incurred by PKs in the presence of TZP, which needs to be elucidated in the future.

Next, AsphyA is clearly demonstrated as a functional protein kinase in mediating plant light signaling using the mutants with reduced or increased kinase activity (Figure 4). Importantly, PIFs are revealed as genuine substrates that are phosphorylated by phytochromes. Thus far, PIF1, PIF3, and PIF4 are shown to be phosphorylated by AsphyA [127], and a recent study demonstrates the regulation of PIF3 and PIF1 by the kinase activity of AsphyA for seedling de-etiolation and germination, respectively [136]. Additionally, since PIF7 and PIF8 are shown to interact with phyA (Figure 2), they can also be regulated by the phyA kinase activity, which needs to be investigated in the future. Moreover, FHY1 and COP1 are also phosphorylated in the nucleus. Thus, it is possible that phyA is involved in the phosphorylation of these signaling components because direct interaction of phyA with them is already reported. Especially, the role of COP1 phosphorylation by phyA is worthwhile to investigate in the future, because HY5 accumulation in transgenic plants with the AsphyA mutants is positively corelated with the kinase activity. There are also other candidate substrates that are phosphorylated by phyA, such as Aux/IAA proteins and cryptochromes. Therefore, the connection or crosstalk between phyA and cryptochromes or auxin/hormone signaling via the kinase activity would be good targets for future studies.

It is also evident that both type I and type II photochroms are autophosphorylating protein kinases. However, the regulatory roles of phosphorylation are not the same among different phytochrome species. For example, while the NTE phosphorylation of phyA induces rapid degradation, that of phyB accelerates dark reversion, i.e., another molecular mechanism of inactivation. In addition, phytochromes interact differently with substrate proteins such as PIFs. For example, PIF2, PIF5, and PIF6 interact with phyB, but not with phyA (Figure 2). PIF1, PIF3, and PIF4 interact with both phyA and phyB, but show different binding affinities. Thus, the regulatory mechanisms of PIFs by different phytochromes during plant growth and development need to be studied further. Moreover, besides PIFs, other substrate proteins that can be regulated by phytochrome kinase activity could also exist in plants. Therefore, it will be necessary to study the functional roles of other phytochromes as protein kinases that phosphorylate substrate proteins in the regulation of plant growth and development.

There are tens of phyA-interacting proteins reported thus far. However, the molecular and regulatory mechanisms of the proteins by phyA are not elucidated fully, although recent studies suggest those of PIFs by phyA (Figure 5). For the regulation of various proteins by a protein, other mechanisms than protein-protein interactions are probably necessary. In this regard, one answer would be the relationship between an enzyme and various substrates. Thus, phyA, as a protein kinase, can regulate various substrate proteins by phosphorylating them, in which the known phyA-interacting proteins are possible candidates of the substrates. In the future, the molecular and regulatory mechanisms of phyA-interacting proteins can be re-examined in view of phyA as a protein kinase. These studies will greatly help to understand the phyA-mediated light signaling in plants.

## Figures and Tables

**Figure 1 ijms-24-02110-f001:**
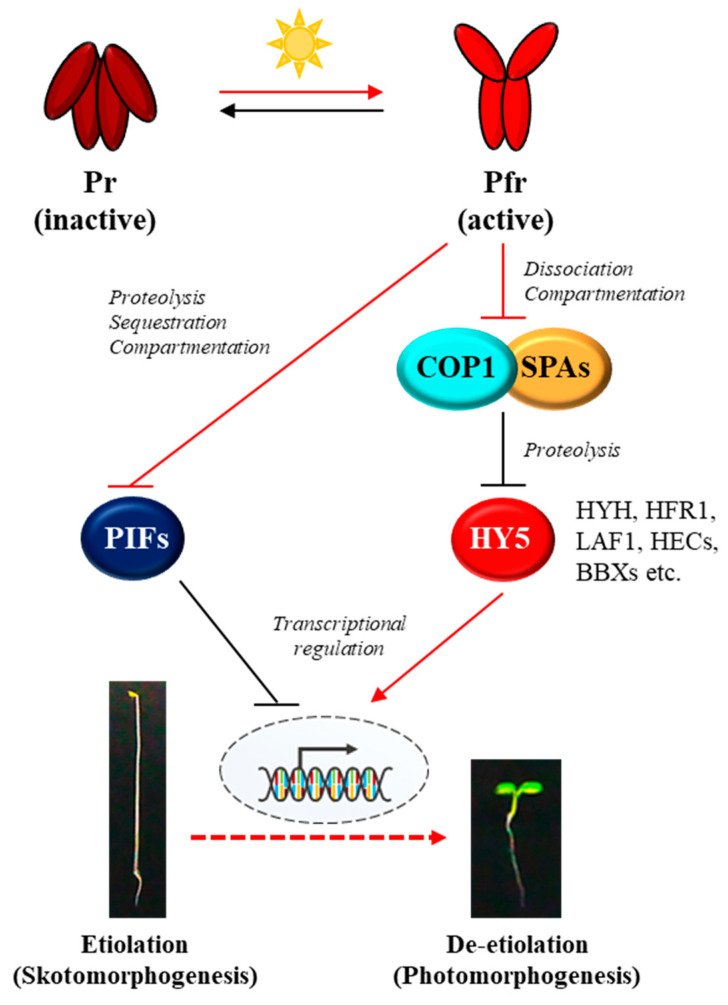
A simplified model of the core phytochrome signaling pathway for photomorphogenesis. Phytochromes are biosynthesized in the cytoplasm as the inactive Pr form in the dark. In the meantime, two repressors for photomorphogenesis, PIFs and COP1/SPA complex promote and sustain skotomorphogenesis by negatively regulating transcriptional factors for photomorphogenesis such as HY5 via proteolysis. Upon exposure to light, the active Pfr form is generated and translocated from the cytoplasm to the nucleus. Then, the photoactivated phytochromes negatively regulate both the PIFs and the COP1/SPA complex. As the regulatory mechanisms for PIFs, phytochromes induce 26S proteasome-mediated degradation (i.e., proteolysis), inhibition of binding to the target promoters (i.e., sequestration), and retention in the cytoplasm (i.e., compartmentation). At the same time, phytochromes induce the dissociation of COP1/SPA complex and the translocation of COP1 from the nucleus to the cytoplasm (i.e., compartmentation), which inactivate the E3-ligase activity of the complex on target proteins. As a result, HY5 is accumulated, which induces the expression of photoresponsive genes for photomorphogenesis. Red arrows and lines indicate the phytochrome-mediated light signaling, while black arrows and lines represent the regulation in the dark for skotomorphogenesis. Pr and Pfr, red and far-red light-absorbing forms of phytochromes; PIFs, phytochrome-interacting factors (8 PIFs in Arabidopsis); COP1, CONSTITUTIVELY PHOTOMORPHOGENIC 1; SPAs, SUPPRESSORs OF *phyA-105* (4 SPAs in Arabidopsis); HY5, ELONGATED HYPOCOTYL 5; HYH, HY5-HOMOLOG; HFR1, LONG HYPOCOTYL IN FAR-RED 1; LAF1, LONG AFTER FAR-RED LIGHT 1; HECs, HECATEs; BBXs, B-box proteins.

**Figure 3 ijms-24-02110-f003:**
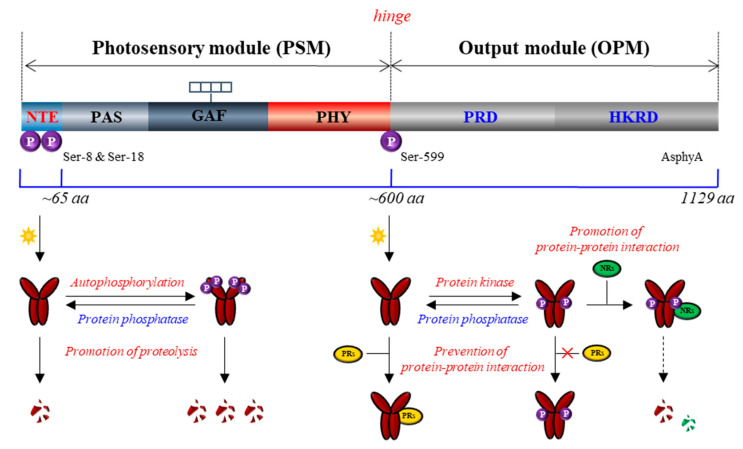
The proposed molecular mechanisms for the regulation of phyA as phosphoproteins. A simplified domain structure of *Avena sativa* phytochrome A (AsphyA; 1,129 aa) is shown. The N-terminal photosensory module (PSM) consists of N-terminal extension (NTE, 1~65 aa) and the photosensory core composed of Per/Arnt/Sim (PAS), cGMP phosphodiesterase/adenylyl cyclase/FhlA (GAF), and phytochrome-specific (PHY) domains. The tetrapyrrole chromophore is covalently attached to a cysteine residue in the GAF domain. The C-terminal output module (OPM) contains a PAS-repeat domain (PRD) and a histidine kinase-related domain (HKRD). The PSM and OPM are linked by a hinge region. There are three phosphorylation sites that have been determined in AsphyA, in which Ser-8 and Ser-18 in the NTE are autophosphorylated and Ser-599 in the hinge region might be phosphorylated by unknown protein kinase(s). The autophosphorylation of AsphyA has been suggested as a mechanism of signal desensitization for terminating phyA function after photoactivation via the promotion of its proteolysis. The phosphorylation at Ser-599 regulates protein-protein interactions with downstream signaling partners, in which the hinge region phosphorylation can prevent its interaction with positive regulators (PRs) such as NDPK2. Based on the results of Arabidopsis phyA phosphorylation site mutants in the hinge region, it is also possible that the hinge region phosphorylation promotes the interaction with negative regulators (NRs) such as PIF3, resulting in the promoted proteolysis of NRs with phosphorylated phyA. This phosphorylation can be reversed by protein phosphatases such as PAPP5. Thus, the phosphorylation in the hinge region can play roles in phyA signaling either negatively (via the prevention of the interaction with PRs) or positively (via the promotion of the interaction with NRs and subsequent proteolysis). The functional roles of the hinge region phosphorylation need to be elucidated more clearly in the future.

**Figure 4 ijms-24-02110-f004:**
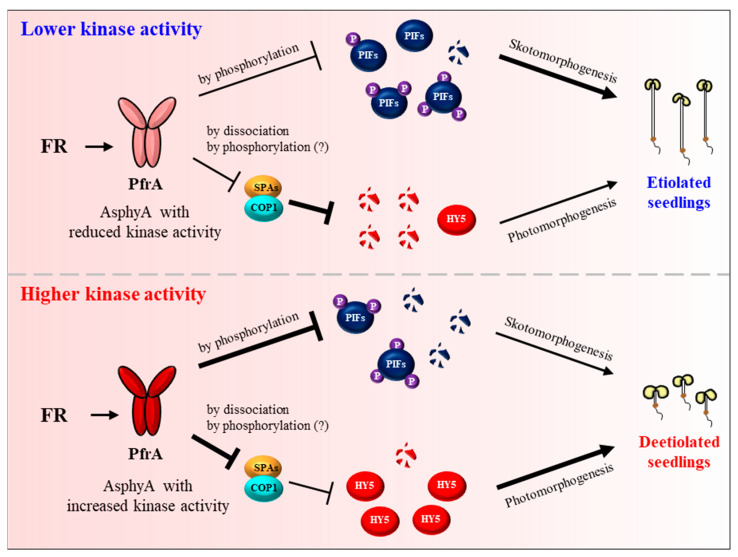
A model depicting phyA signaling as a protein kinase in response to FR light. This model is constructed using the results of AsphyA mutants with lower or higher kinase activity. Based on the core phytochrome signaling pathway, far-red (FR) light-activated phyA (PfrA) phosphorylates PIFs (labeled ⓟ in purple), which induces subsequent degradation via the ubiquitin/26s proteasome pathway. At the same time, PfrA induces the dissociation of COP1/SPA complex, resulting in the accumulation of HY5. Thus, the negative regulation of PIFs and COP1/SPA complex by phyA resultantly contributes to HY5 accumulation, which is necessary for photomorphogenic development such as de-etiolation of seedlings. According to this model, AsphyA with reduced kinase activity might exhibit an attenuated function (i.e., approximately 2-fold decreases in phosphorylation of PIFs and dissociation of the COP1/SPA complex, represented as thin lines), showing reduced responses of transgenic plants to FR light (i.e., etiolated seedlings). On the other hand, AsphyA with increased kinase activity could show an enhanced function, which results in about 4-fold higher accumulation of HY5 (represented as thick lines), exhibiting increased FR-responses of transgenic plants (i.e., de-etiolated seedlings). It is noted that other protein kinases are also involved in the phosphorylation of PIFs, and possibly PfrA is involved in the phosphorylation of COP1 (see the text).

**Figure 5 ijms-24-02110-f005:**
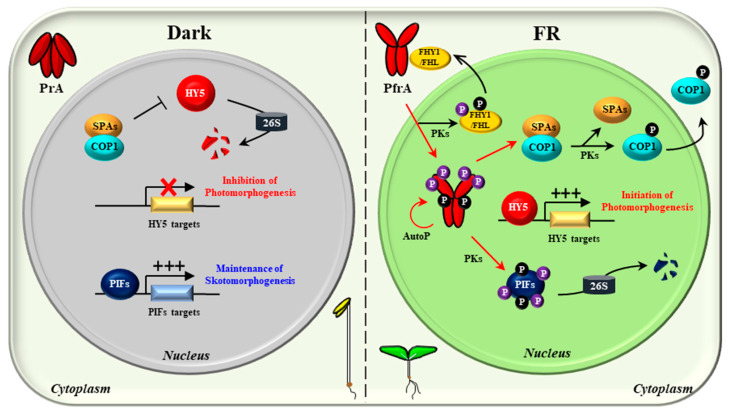
A model for phyA-mediated plant light signaling as a protein kinase. In the dark (left), phyA is biosynthesized as the Pr form (PrA) and accumulates in the cytoplasm. PIFs accumulated in the nucleus regulate transcription of genes to sustain skotomorphogenesis. At the same time, the COP1/SPA complex degrades HY5 via the ubiquitin/26S proteasome pathway to prevent photomorphogenesis. Under FR light (right), phyA is photoactivated to the Pfr form (PfrA) and translocalized into the nucleus using FHY1/FHL shuttle. As a protein kinase, PfrA phosphorylates PIFs to induce the degradation of PIFs, and inactivates COP1/SPAs by inducing the dissociation of the complex. As a result, HY5 is accumulated and induces the transcription of photoresponsive genes to initiate photomorphogenesis. In addition, autophosphorylation of phyA (AutoP) occurs for signal desensitization by inducing rapid degradation of phyA. Red arrows represent the molecular mechanisms of phyA to induce photomorphogenesis (labeled ⓟ in purple), while black arrows represent the generally known mechanisms in previous studies. Besides phyA, other protein kinases (PKs) are also possible to involve in the phosphorylation of phyA (hinge region), FHY1/FHL, COP1, and PIFs (labeled ⓟ in black).

## Data Availability

Not applicable.

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
