# Peer review of "Regulation of Plant Photoresponses by Protein Kinase Activity of Phytochrome A"

_ijms, 2023, doi:10.3390/ijms24032110_

Round 1

Reviewer 1 Report

The present manuscript entitled "Regulation of Plant Photoresponses by Protein Kinase Activity 2 of Phytochrome A" is a well designed and well planned work. It can be accepted for publication after few minor changes.

1. If possible try to add a table which constitute with different photo-responsive genes found in different plants (and specific organs) and their relation with phytochrome and downstream action. 

2. Separate Conclusion and Future prospect into two sub sections.

3. Use author citation during first mention of scientific name.

4. Check typos.

5. Check reference style. 

Reviewer 2 Report

The manuscript "Regulation of Plant Photoresponses by Protein Kinase Activity of Phytochrome A" by Choi et al. is an interesting, logically composed review of the existing evidence on the mechanism through which phytochromes pass on red/far red light signals in order to regulate plant morphology. The diagrammes provided by the authors are clear and are in line with the text (they support each other).

Contentwise, I have no questions. The authors have performed an extensive review of the literature, adding to the work of the recent large review by Cheng et al. (also referenced in this work).

Languagewise, there are a few minor comments to address:

- line 30, "which makes to evolve" should be "which caused the evoution of multiple..."

- line 44: THROUGH exposure to light

- line 46: "Due to the difference in light-dependent protein stability..." difference between what and what ? phyA and type II phytochromes ?

- line 98-101, more specifically "have been known as", change to "have been identified as", or even better "Phytochrome mediated regulation occurs through..."

- line 122: ...negatively regulate BOTH THE PIFs and THE COP1/SPA complex"

- line 123 ... induce 26S proteasome-mediated degradation...

- line 134: Phytochromes ARE INVOLVED IN...

- line 196: ... the hinge region phosphorylation plays A ROLE in the regulation...

- line 254-255: ...(OPM) containS A PAS-repeat domain and A histidine kinase-related domain...
